# Edible Garden Cities: Rethinking Boundaries and Integrating Hedges into Scalable Urban Food Systems

David Adams [1],*, Peter J. Larkham [2] and Michael Hardman [3]

1   School of Geography, Earth and Environmental Sciences, University of Birmingham, Edgbaston, Birmingham B15 2TT, UK
2   College of the Built Environment, Birmingham City University, Millennium Point, Curzon Street, Birmingham B4 7XG, UK; peter.larkham@bcu.ac.uk
3   School of Science Engineering and Environment, University of Salford, Salford M5 4WT, UK; m.hardman@salford.ac.uk
*   Correspondence: d.adams.4@bham.ac.uk

**Abstract:** Connecting to and extending recent debates around more-than-human thinking, this paper explores how porous boundary treatments and plot layouts might encourage ecological exchanges within new urban and peri-urban developments. This study therefore responds to suggestions for innovative plot designs that facilitate positive trans-species interactions, especially considering wider anxieties surrounding biodiversity loss and recognition of the need for climate-resilient garden spaces. Focusing on a recent example of a large-scale residential development in the English midlands, this paper outlines the socio-economic, cultural and ecological significance of embedding different hedgerow designs into early planning considerations; revealing the need to move beyond current models. The discussion then turns to how such ambitions might encourage sustainable land use, particularly through creating potentially scalable urban agricultural systems that sustain healthy food choices.

**Keywords:** hedges; boundaries; urban food systems; suburban design; England; planning

## 1. Introduction

Parts of the social sciences have recently given much attention to how urban space is constituted through, and produced by, a range of human and non-human 'actors'. Studies explore the diverse ways in which revitalized public open spaces, remediated land, urban farms, restored urban watercourses, and the creation of green corridors, alongside other forms of carefully managed 'green urbanism', have inflated property values in contiguous city-centre neighbourhoods, resulting in the exclusion/displacement of businesses and residents (for example, [1]). One outgrowth of this work involves an examination of how certain animals and plant life are either 'vilified' and/or 'celebrated' in plans that add economic value to property [2]. Indeed, investigating the diverse ways in which certain animals and plant species constitute active, 'lively' resources helps to soften dominant 'hylomorphic' intrusions into urban landscape. Developing this perspective can inform planning processes and design interventions which better acknowledge how different species align with, evade or directly challenge human-centred models of contemporary urban renewal that emphasize the importance economic exchange in facilitating development [3–5].

Inspired by recent accounts suggesting building more compassionate planning decision making [4], this study examines the durability of historically rooted suburban notions of domesticity reflected in boundary designs. Such thinking continues to influence contemporary residential developments at the peri-urban fringe that mediate public–private interactions and permit and/or downplay positive human–nature connections [6]. Generating possible inventive plot and permeable boundary treatments that might facilitate

constructive 'trans-species' exchanges at the urban–rural edge remains an important endeavour. This is especially significant, given broader concerns regarding development pressure at the urban-rural fringe [7], biodiversity loss and preparing gardens for the shifting impacts of climate change [8]. In this context, several studies recommend the use of mixed-species hedges in residential contexts [9–13]. These can provide 'natural' pest control, shelter, food, carbon storage, infiltration promotion, soil nutrients, and increase insect pollinator and invertebrate diversity [9–11] [1]. Hedges also act as air pollution barriers and windbreaks and can mitigate issues around absorbing/reducing particulate matter, while also creating aesthetically pleasing boundaries for food growing [9,13].

Although many recent accounts have focused on hedgerows and the aesthetic and biodiversity qualities in existing garden spaces, this paper examines the potential for hedges to be used for productive purposes, from the direct incorporation of growing, or as part of the protection and development of peri-urban food growing spaces within new edge-of-settlement residential schemes. This extends the work that has emerged in recent decades that calls for stronger planning and design instruments that generate potentially scalable models of agricultural-led peri-urban development, and which enhance liveability in a climate-changed future [14–16].

Using a 'live' example of a large-scale residential development in the English midlands, this paper repurposes the adopted scheme and sets out a reworked masterplan for the site. This considers the importance of embedding existing and newly planted hedgerows—as spontaneous or planted structures of trees, shrubs and fruit-yielding species—into early design thinking of new residential layouts which connect historical landscape features, cultural heritage and hence generate potentially scalable urban food systems [16]. This design approach is potentially replicable in other development contexts.

## 2. Suburbanizing Nature

As with the use of earth banks, ditches, and wooden palisade fencing, the planting of hedges represented early efforts to enclose nature: while they were often built for practical reasons, these structures often radiated messages of legitimacy, privacy, safety and ownership, thus helping to keep danger, disease and peril at bay [17]. In the medieval period, before the enclosure movement, hedges provided shelter for livestock, served as boundary markers as well as a source of food, timber and fruit [11,12]. Centuries later, planted hedges, for example, became indelibly linked with landscape aesthetics and the protection of wealthy landowners, resulting in the dispossession of common land and rights through legislative enclosure, compelling rural labourers, especially in the nineteenth century, to seek out opportunities in rapidly expanding towns and cities [17]. A revulsion against the subsequent social, political, and economic upheaval, and unhealthy living conditions of urban centres, sparked municipal governments' public health interventions to create 'deodorized' and civilized living environments [2]. The burgeoning urban middle classes, seeking more stable and wholesome living environments, found refuge in those mainly single-family and privately-owned garden suburbs, replete with hedge-fringed gardens, built during the early-to-mid-twentieth century [10].

The relatively unregulated suburban growth of the early-to-mid-twentieth century provided a bulwark of sorts against a diverse array of urban threats; houses and plots were thus infused with themes of family life, health, privacy, safety, and social conformity [18]. Although there were some notable, albeit piecemeal, efforts to reverse these social and design tendencies in England, with planned estates fronting onto green public open spaces [19], the widespread use of privet and yew hedges tended to act as defensive boundary markers; 'unsightly' fencing could be beautified by 'appropriate planting' of carefully selected flowers, shrubs, espaliers and trees to add colour, texture and life [20]. Rear gardens, for example, served as light and airy 'outdoor rooms', and were typically expected to be civilized, private spaces, reserved for pets, children's play, and the controlled growing of decorous flowers [20]. While home produce was championed during times of economic uncertainty and during the First and Second World Wars, the allocation of

garden space for rearing livestock or vegetable growing was largely incompatible with messages circulating among some architects and the popular garden press, which stressed the value of having allotments and other community growing spaces situated away from the domestic sphere [21].

The subsequent imposition of planning controls on land use, and green belt protection after the Second World War resulted in a general urban shift in parts of England and continental Europe away from the expanding fringes of existing settlements and towards government-sponsored new towns and denser urban living in towns and cities. Likewise, increases in overall levels of post-war prosperity, shifting personal mobility patterns, leisure habits and diets, have also resulted in a reduction in allotments, hedged gardens, and spaces dedicated to household food production [21]. For decades, the location and design of new housing areas have led to conflicts with long-established landscape features, wildlife conservation and agricultural productivity. In the commuter belt north of London in the 1970s and 1980s, "a landscape of trees and hedgerows hiding large houses started to be chopped back to accommodate smaller, unambiguously urban dwelling types" [22], with planning decisions encouraging certain species, and prioritizing standard designs and infrastructures, instead of potentially unsettling, 'out-of-place', outmoded agricultural practices, and uncivilized aspects of nature associated with the pre-suburbanization environment. These factors, combined with a rise in over-engineered field boundary treatments [17], and increased reliance on international food supply chains to serve increasingly diverse urban populations, have resulted in a general decline in hedgerows, despite sustained conservation efforts designed to halt their extensive removal.

Whatever associations hedges and hedgerows might have acquired in terms of their historical role in the curtailment of rights, maintenance of elite privilege, or as a symbol of repressive suburban nostalgia [22], recent attention focuses on developing a more fluid view of the value they can play in contemporary society. Rather than a solid barrier, or a stable motif of conservation, hedges have come to be regarded as powerful marshalling points for a range of contemporary environmental concerns. Several recent accounts explore the possibilities and limitations of reintroducing and extending hedgerow networks across diverse urban tapestries [9–13]. The focus of these studies is on assessing how different hedge types, species, and 'time-tested' hedging techniques might serve as valuable 'nature-based solutions' [11], shaping contemporary urban land use systems and practices [12].

Of course, creating and/or retrofitting vibrant hedgerow networks across heavily-fragmented urban environments, with different land uses, property ownership boundaries and unsuitable surfaces, remains fraught with practical difficulties [12]. Maintaining hedges can also take considerable time, craft, effort, and financial outlay; excessively high hedges also block light, drink too greedily from the soil and spark inflammatory neighbour disputes [17].

However, given the recent decline in vegetative cover across fragmented urban landscapes, and enduring concerns and national news stories about the unsympathetic removal and/or management of hedges and habitats [23], further work is needed that recognizes the significance of these "under-appreciated assets" [10]. This is especially important given wider calls for planning frameworks and design interventions that encourage biodiversity by creating resilient gardens in readiness for the growing impacts of climate change [8]. And this is thrown into sharp focus, given the development pressures being experienced at the expanding penumbra of urban settlements, where demand for new housing is high and human–nature exchanges are arguably most pronounced [14].

## 3. Ecological, Biodiverse and More-Than-Human Issues

Many official planning processes and design interventions are coming to recognize the significance of more-than-human agency [3,4]. For example, wildlife corridors, incorporating fruit-bearing trees, shrubs and hedgerows, log-piles for microfauna, nesting boxes for indigenous bats and birds of prey, retained/restored wetlands for reptiles and amphibians, hedgehog houses, insect hotels, pet-friendly infrastructures and so on have

recently become intimately woven into planning discourses and the marketing of new developments [5] [2]. Localized nature-based initiatives promote the value of different urban green spaces—including hedgerows and urban food growing initiatives—that deliver a range of ecological, social, physiological, physical and emotional benefits [24]—more widely valued since COVID-19. These and other efforts doubtless add vitality to new large-scale developments that are routinely criticized for creating ostensibly sprawling car-orientated, 'placeless' and nature-depleted monofunctional middle-class estates [6,25].

Designs for large-scale residential developments often include health, education, community, retail, transport and leisure facilities, and a mixing of tenures. Similarly, a mix of footpaths, cycleways, ponds, play areas, and allotments is reflected in green infrastructures. Cumulatively, these can encourage healthy lifestyles, socialization and inclusion. But, even leaving aside the more general criticisms surrounding whether new developments are sufficiently served by local amenities [25], certain design ambitions tend to rest on a selective biophilia, with wildlife being regarded as uneasy/difficult-to-manage intrusions in human-created urban spaces. Statutorily 'protected' habitats, 'healthy' trees, hedgerows and animals are prevented from being displaced during processes of urban development; vegetation, too, is subject to control, permitted to flourish under certain conditions and in regulated spaces. 'Unruly' and/or 'unwanted' plants, trees and hedges are routinely pruned, trimmed or removed, despite recent gardening styles and campaigns that encourage a degree of managed disorder [26]. Residents may choose pollinator-friendly seasonal plants, and keen gardeners will embark on redesigning their plots, incorporating wildlife-friendly trees, leaving areas to be 'unkempt'; they may add water sources and increase composting [8].

But, sustaining these ambitions will also vary among different individuals ranging from wholesome neighbourly interactions, concerns for wider environmental issues, animal welfare and sustainable forms of food production, and support for vibrant ecologically rich gardens and local sites, to careless abandonment and unneighbourly attempts to protect property from real and imagined threats. Likewise, pets and 'fuzzy' wildlife (garden birds, bees, butterflies, and hedgehogs, for example) are often welcomed as 'charismatic' and/or companionable, and thus valued in garden and neighbourhood spaces. An increase in dog ownership among some residents, for example, may be read as being emblematic of socio-economic status and serves as a symbol of harmonious family life [2]. Yet, these notions are also undercut by recent concerns over the potentially recalcitrant behaviour of attention-starved dogs, especially as residents either returned to office-based work or became increasingly engaged with teleconferencing calls and other homeworking activities: such anxieties have precipitated a growth in professional dog-walkers [27]. Hence unwanted, boisterous and/or disease-carrying animals, pathogenic water sources, invasive flora and so on are typically categorized as nuisances or pests. Their presence threatens the innate human need for safety, comfort, privacy, and expression [6], and thus becomes subject to eliminative policy discourses, defensive architectures, control and even extermination mechanisms that reinforce human-centred capital and property interests [2].

These human–nature tensions are prescient and relevant in the context of new residential layouts and the design of boundary treatments. While new housing developments will include manageable landscaping, schemes are typically characterized by impermeable surfaces, with plots having extensive paving, turf and/or bare earth often surrounded by impervious paving, walls and fences. Recently, even the 'turf' may be artificial. This includes the relatively recent roll-out of 6 ft × 6 ft industry-standard, pre-treated softwood closeboard or panel fencing. Typically, such fencing can be up to two metres high without planning permission; and these panels are often constructed off-site from 'custody certified' sustainably sourced timber and to accord with developers' trade specifications (Figure 1). These comparatively durable, low-maintenance and commonplace boundary fences undoubtedly create saleable, 'safe' and easy-to-care-for plots coveted by certain professional classes, especially those seeking flexible live–work patterns, and which remain within touching distance of urban and rural amenities. Nonetheless, such boundary treat-

ments tend to revive suburban anxieties regarding the need to control or eliminate those uncontainable and threatening aspects of nature, while also inhibiting the number, type and movement of different flora and fauna, including the under-threat European hedgehog [8]. The cumulative impact of such a design thinking could have dire consequences, particularly given the desire to upscale these developments to meet population demands and to enable a more equitable housing market.

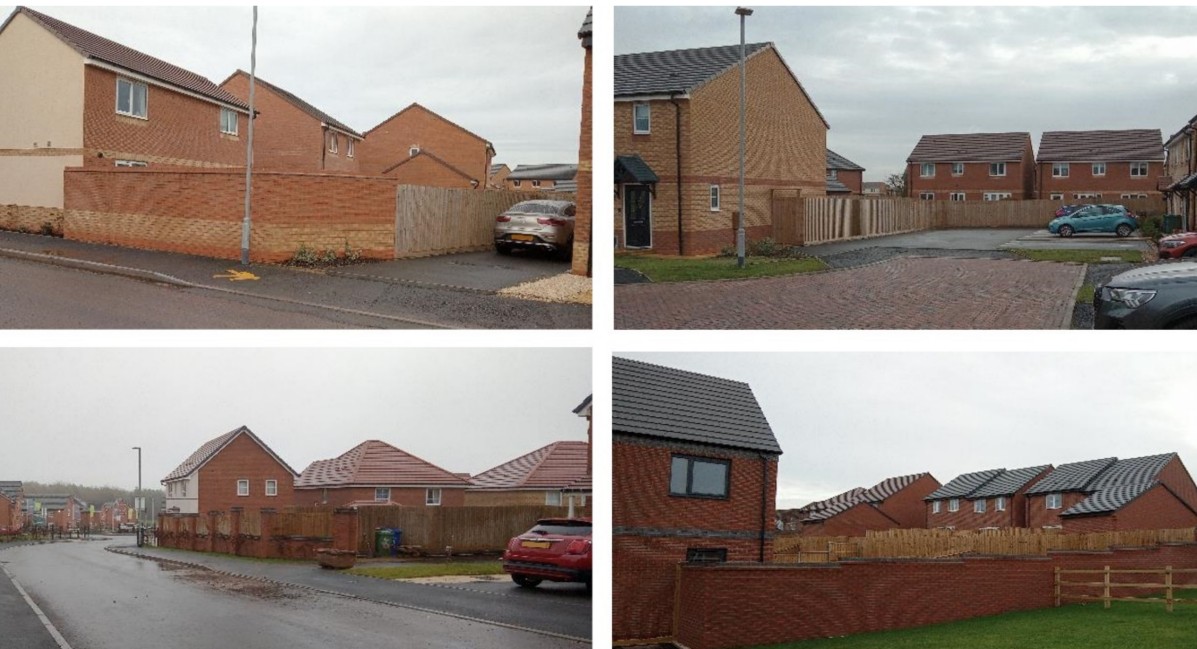

**Figure 1.** Examples of 'hard' boundaries in recently built edge-of-settlement residential schemes. Source: Authors' own photographs.

A degree of caution is needed here. Despite the seemingly impenetrable boundaries and surfaces, analyses of the urban spatialities of foxes, badgers, rats, flies and cockroaches across different geographical contexts suggest that some species resist, defy and thrive, despite residents' attempts to erect boundaries [2]. Even 'secure' residential garden boundaries are routinely breached by an uncontrollable array of sights, sounds, and smells. The relatively unhindered movement of other residents' pet cats may disturb human efforts to define and protect boundaries, prompting a range of emotive human responses to 'wild' nature; some residents with a keen interest in encouraging garden birds may strive to deter unsolicited, predatory feline incursions [28].

Fences and walls may be 'reclaimed' by a multitude of birds, plants, and insects. Boundaries will decay, 'fail', or be replaced, perhaps with more 'permeable' options, including hedges: residents will alter 'their' plots to suit the shifting vicissitudes of taste and circumstances. Of course, few would argue that animals and plant life which either spread disease or severely disrupt gardens and household life should have inalienable rights, specifically in those circumstances where residents are vulnerable to infectious disease. Yet, Hubbard and Brooks [2] note that, in working towards developing more-than-human planning frameworks, based on ecological rather than economic exchange value and ownership, attention should also focus on developing implementable planning mechanisms that can support a middle way between the imposition of human will on the environment and letting nature take its own course. Reconsidering the practical elements of garden design, including the role played by the seemingly overlooked boundary treatments, is a necessary step towards this ambition.

## 4. Growing Ambitions

Exploring the opportunities of integrating 'porous' boundaries at the masterplanning stage of large-scale residential development. It offers one practical response to recent calls for compassionate forms of planning and design [4] and actions that acknowledge "trans-species" forms of co-existence [3]. Many innovative urban design and planning examples emerging in parts of continental Europe, Asia, north Africa and the 'shrinking cities' of the United States highlight possible ways to link rural food production and urban consumption, shorten supply chains, and generate networked peri-urban cultural landscapes [14,15]. Encouraging and much-discussed models exist that seek to achieve these goals, including concepts such as Continuous Productive Urban Landscapes, and Edible Green Infrastructures [16]. Here, for example, land zoning, planning ordinances, and inventive forms of land tenure can help deliver 'recreational facilities, climate adaptation' [15] and encourage biodiverse, liveable environments.

Rather than having development proposals that view forms of life as being expendable and/or 'worthy' of special protection and hence marketable, certain designs tend to emerge from existing ecological conditions. Chen [9], for example, provides an analysis of how edible hedgerows and other ecological features in the rural landscape of the US could act as a catalyst for sustainable design thinking. Other studies demonstrate and explore the strengths and weaknesses of creating residential layouts based around existing agricultural activities, underpinned by a communitarian spirit and wide-ranging ecological conservation practices (for example, refs. [29–31]).

These schemes embody broader ambitions to 're-localize' food systems, reduce a reliance on expensive fruit and vegetables, and create productive urban/peri-urban landscapes, linking food production with other regional infrastructure [14]. This includes a recognition of the everyday nature of hedged fields and communal growing spaces; these are then used to structure plot design and street layout. Such an intervention is often designed to tackle issues around perceived food deserts, reconnecting urban residents with produce in a more meaningful manner. Inevitably, these developments raise the spectre of low-density, unsustainable expansive suburbs designed around human liveability, capital accumulation and the desires of narrowly defined socio-economic groups. Without a shared, implementable vision to bind together relevant stakeholders and supportive planning instruments [14,32], these schemes may offer an artificial version of a healthy, farm-fresh lifestyle, permitting highly managed, commodifiable human–nature exchanges. Yet evidence provided from developers, farmers, planners and local residents suggests that well-designed and actionable models of equitable agricultural production can succeed in improving residents' health, creating important sites based on social and ecological exchange among diverse communities [30].

Nevertheless, despite Chen's US-based account [9], few, if any, studies have provided a careful analysis of existing and new hedgerow networks to guide the design of new residential layouts capable of supporting agricultural models of development in the UK. But, this task is particularly pressing, given both the need to provide appropriately located sustainable and affordable housing [7], but also because there is a demand to create healthy, resilient food systems, investment, employment and training opportunities, while delivering environmental benefits [14–16]. Moreover, alongside their environmental qualities, such developments carry the potential to add much-needed texture, temporal depth and ecological character to those seemingly nature despoiling, characterless/'placeless' sites, typically associated with single-family suburban households with sedentary, unhealthy lifestyles [6]. Alongside the direct potential for incorporating produce into such spaces, few studies examine the wider value of these assets in protecting existing or new community food growing spaces, potentially enhancing their social, environmental, health and economic value.

## 5. Methods

Within this study, the reviewing and identifying data related to gardening and hedgerows draws on primary observational data and secondary material related to the evolution of gardens and hedgerows. First, the paper examines how innovative boundary treatments as potentially scalable design features might be embedded in a real-life new-build developments. A case study approach was used, which focused on one large-scale strategic urban extension in Stafford, some 40 km north of Birmingham. This case typifies recent market-led approaches to housing delivery and planning approaches which tend to direct development at the fringes of an established settlement (Figure 2), thus enabling potential replication on a wider scale, particularly across the UK. Strategically, the site was enshrined in the area's local plan as a "sustainable, well designed, mixed-used development" [33], and one that "builds on its inherent assets, its existing topography [and] ecology" [34]. The site comprises several tracts of managed grassland, fallow fields and land set aside for pastoral farming. Various planning permissions were secured for the phased creation of over 1000 houses, elderly living facilities, primary and secondary schools, a local centre, and green infrastructure.

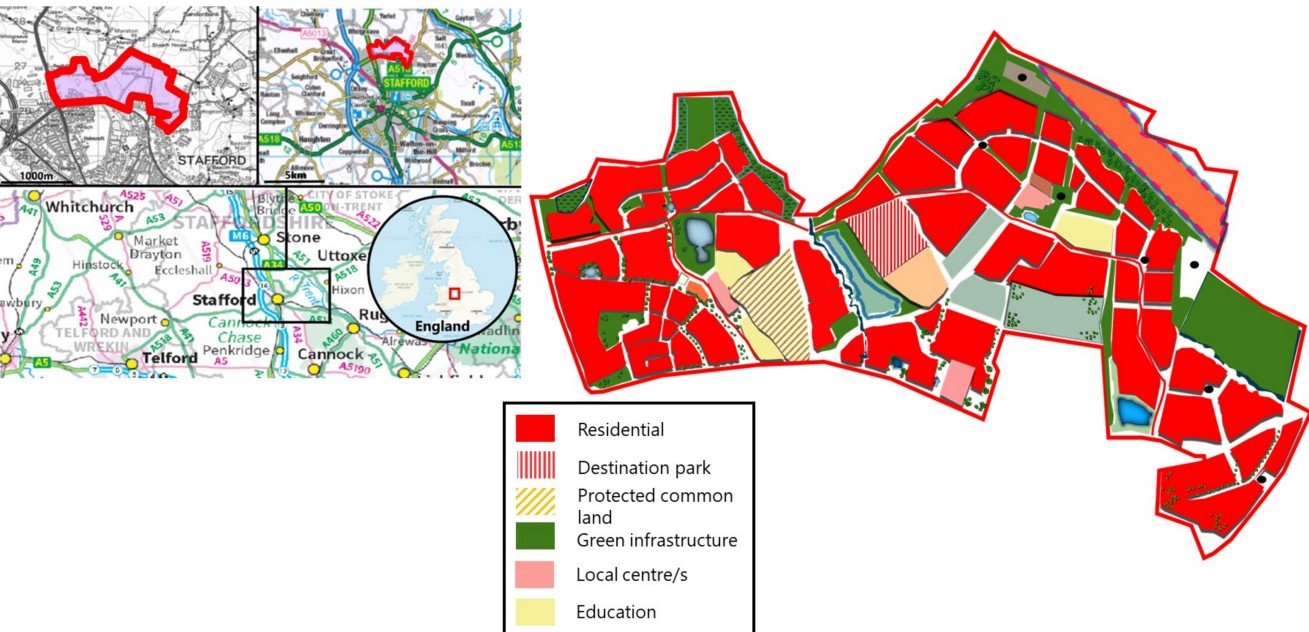

**Figure 2.** Location of the North of Stafford Masterplan (**left**). Source: Contains OS data © Crown copyright and database rights 2023 Ordnance Survey. North of Stafford Masterplan (**right**) [34].

Although much of the site is effectively 'built out', lessons can be taken from reviewing the planning discourse and decision making regarding the preservation and integration of hedgerows to strengthen local character, while also examining proposals for their ability to encourage biodiversity, build climate resilience and facilitate positive human and non-human exchanges. Specifically, using the schema developed by Collier [11] to test the effectiveness of nature-based solutions, designs were given a score ranging from 1 to 5 for each category; these scores were then aggregated. Designs were chosen by reviewing and collating different boundary options; these were taken from popular online home-improvement/gardening resources, and from Ripani's typology of "living fences"—a collection of fences "made using plants on their own or by combining plants with appropriate structures" [35]. Many of these options appeal to residents wishing to retrospectively improve their plots from the minimum-cost standardized boundaries with which houses are sold. However, few, if any, studies have assessed the potential of these designs to provide cost effective, replicable, innovative and implementable boundaries at the early planning stages of new-build development.

These ideas extend those accounts which outline the need to develop actionable trans-species urban greening efforts by reviewing recent efforts to build sustainable peri-urban models [14,15]. This analysis was supplemented with an analysis of the possible benefits attached to hedgerows as everyday landscape features which connect history, ecology and sense of place [9] in ways that might support the design and implementation of urban hedgerows as valuable everyday features in wider sustainability debates. In addition, detailed analysis was performed on this development; this included (i) an analysis of site areas and boundary conditions, (ii) a consideration of the relevant planning history relating to existing green infrastructure, including networks of hedgerows, (iii) identifying any potential benefits that may be created by developing improved hedge designs, and iv) identifying those plots and locations within the site which offer potential opportunities for plantation and food production, taking into account the historical hedgerow network and the different nature and type of boundary design. Informed by ideas regarding the potential services and benefits of transferring hedgerow systems into urban contexts, a selection of boundary treatments was scored against different categories [11,12].

Ultimately, this exercise presents one framework for embedding ecological and 'more-than-human' approaches into scalable land use decisions, particularly around food growing to deliver social, health and other therapeutic benefits. In doing so, this case study demonstrates how such an approach can be replicated at a wider scale to understand these issues within a broader context.

## 6. Pushing Boundaries

Rethinking and reshaping the design ambitions associated with the Marston Grange development offers an opportunity to consider how to embed hedgerow ideas into new developments. Land at the Marston Grange site was acquired by Azko Nobel UK Ltd. from Courtaulds during the late 1990s and subsequently earmarked as a strategic development site in Stafford Borough Council's local plan. Volume housebuilders, including Barratt West Midlands and Bovis Homes, then took on the project, and extensive public consultation was undertaken [36]. This included the distribution of some 11,500 leaflets delivered to local homes which informed local communities about the concept plan; emails were sent to local community, voluntary and third sector groups; a public exhibition and workshops were held with local councillors and school children [36]. This evidence gathering highlighted pockets of praise directed at the housing design and general concept of development. However, documented concerns included the potential disruption, pollution and noise created by additional traffic, a perceived high-density development with insufficient build quality, unsatisfactory amenities, and lack of affordable houses: other issues were raised over the possible impacts on wildlife, flooding and "agricultural land/food security" in an area historically associated with arable and pastoral farming [36].

This evidence was considered by the local authority following formal engagement with councillors and statutory consultees, and designs were duly modified. Indeed, the masterplan included multi-use play spaces and leisure facilities to encourage "happy and healthy" living, flood retention schemes, green corridors, and the identification of allotments, thus building a "strong landscape character" [34]. Furthermore, given longstanding concerns surround established market-driven models of housing development which focus on 'profits and quantity' [25], these initiatives are also suffused with established discourses of ecological restoration, which embed marketable forms of wildlife to enhance attractiveness of development [5]. For example, local habitat and ecological surveys reported the aboricultural, landscape and conservational value of statutorily protected and endangered species found across the site: reptiles, badgers, roosting bats, and breeding birds, amphibians, grassland habitats, watercourses and ponds, mature trees and hedgerows were all recorded [37,38]. These findings were then reflected in design ambitions regarding the "potential for habitat creation, including new tree and shrub planting along with the new ponds" and the provision of "additional detailed enhancements, such as installation of bird and bat boxes" [34].

Although the masterplan recommended that field patterns be "retained where possible", no explicit mention is given to retaining or enhancing "low grade agricultural land" [38] used for arable farming; this suggests a subtle politics of displacement/eradication at work. Similarly, developers were also requested to demonstrate how "the biodiversity value of the site will be enhanced", through the retention and enhancement (where possible) of "trees, hedgerows and ponds" [39]. Yet, while this occurred during the building phase, analysis of aerial photographs also reveals considerable removal of hedges and mature trees, despite suggestions made to "replant a wide range of species suited [...] to the landscape setting", such as "limes [that] have been used to develop this part of Stafford" [37]. Of course, the selective removal, displacement and management of existing flora and fauna is often justified on economic and practical grounds [23]. But, despite concerted efforts to conserve rural hedgerows as "icons of the English rural aesthetic" [40], together with increased awareness of hedges to improve carbon capture and halt biodiversity decline [13], this development extends a trend of tree and hedgerow removal affecting other parts of England in recent decades [17] (Figure 3).

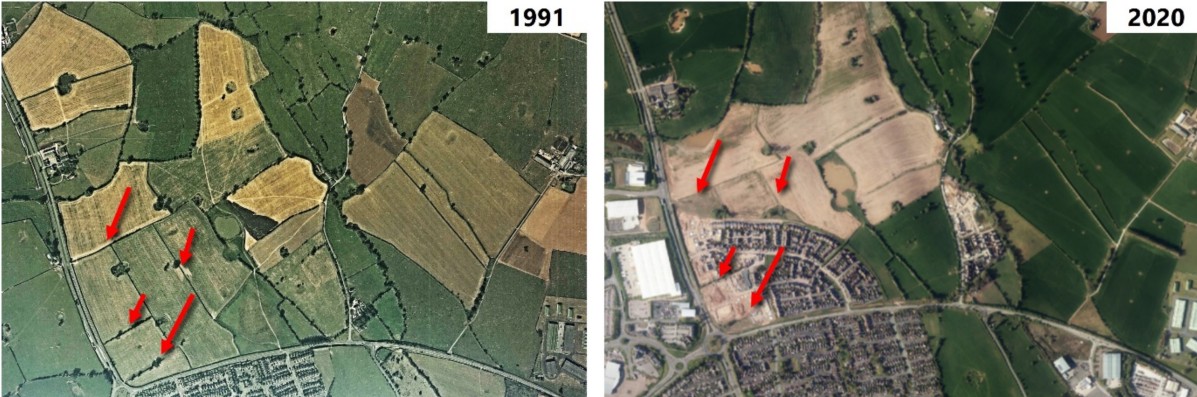

**Figure 3.** Aerial images showing the hedgerows running across the Masterplan site in 1991 and 2020. The arrows show existing hedgerows (**left**), and the early stages of construction (in 2020) (**right**). This shows the loss and/or modification of existing hedgerows. Aerial photographs obtained from Staffordshire Record Office. Map contains OS data © Crown copyright and database rights 2023 Ordnance Survey.

The existing scheme also makes provision for pedestrian movement, improved public transport routes and alternative modes of travel [34]. However, historical ideas around privacy, individualism and security are also firmly embedded in the official design ambitions, with a minimum of "two car parking spaces" being allocated per house, while "enclosed rear gardens" and "timber fencing [for] plot boundaries" are encouraged [33]. Such use of such fencing arguably reflects a suburban tradition; they succeed in creating safe, secure, private and healthy back garden 'havens' [20], allowing residents a space for individual expression in ways that suit residents' shifting needs, tastes and circumstances (Figure 4).

While this may be the case, alternatives to 'industry-standard' pre-treated wooden fencing exist; these often include a mix of wood and natural fibres, recycled/composite plastics, metal, stone, hedges, and native plants, trees and shrubs are often presented as potential options in this regard [35]. Against a broader context of integrating sustainability in garden spaces, residents of established residential plots may choose inexpensive, low-maintenance, attractively-designed and practicable, permeable boundary solutions. These undoubtedly appeal to certain residents, especially those with the motivation, time, money and resources and a degree of 'outward-looking', neighbourly cooperation, all of which are needed to retrofit gardens to include wildlife-welcoming, permeable modifications. But local social media reports tend to confirm residents' desires to manipulate their local environments in ways that maintain orderly, peaceful neighbourhoods, comprising compliant pets as 'living property' [2] that fit with settled, acceptable notions of domestic life.

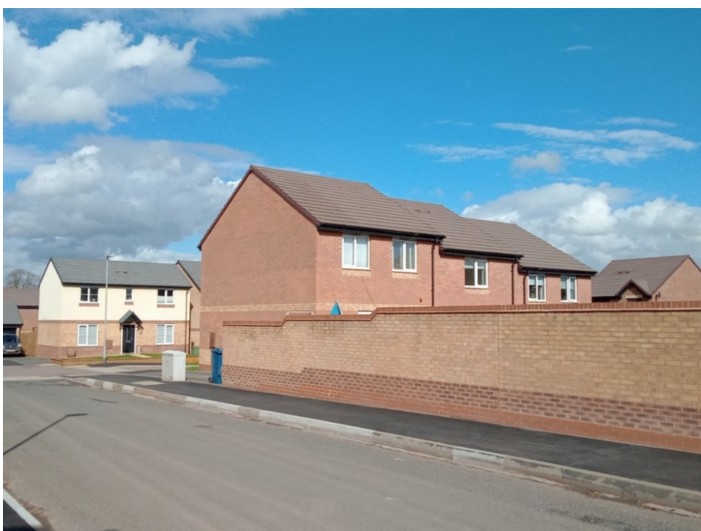

**Figure 4.** 'Hard' landscaping at the Marston Grange development. Source: Authors' own photograph.

These ideas are reflected in requests made via social media for local dog-walking services, gutter maintenance, local cleaners and garden-decking/slabbing services. Other stories report uncooperative attempts at hedge trimming, complaints regarding unidentified and noisy cats wilfully contravening property boundaries and, in some cases, defiling neighbours' unopened milk bottles (Figure 5). Such accounts raise concerns over the threat of truculent and/or insouciant 'lower status' resident behaviour; if left unsupervised, these and similar anxieties would likely breach domestic boundaries, leading fears about public health, the transmission of potential diseases, and a disturbance of certain 'norms'.

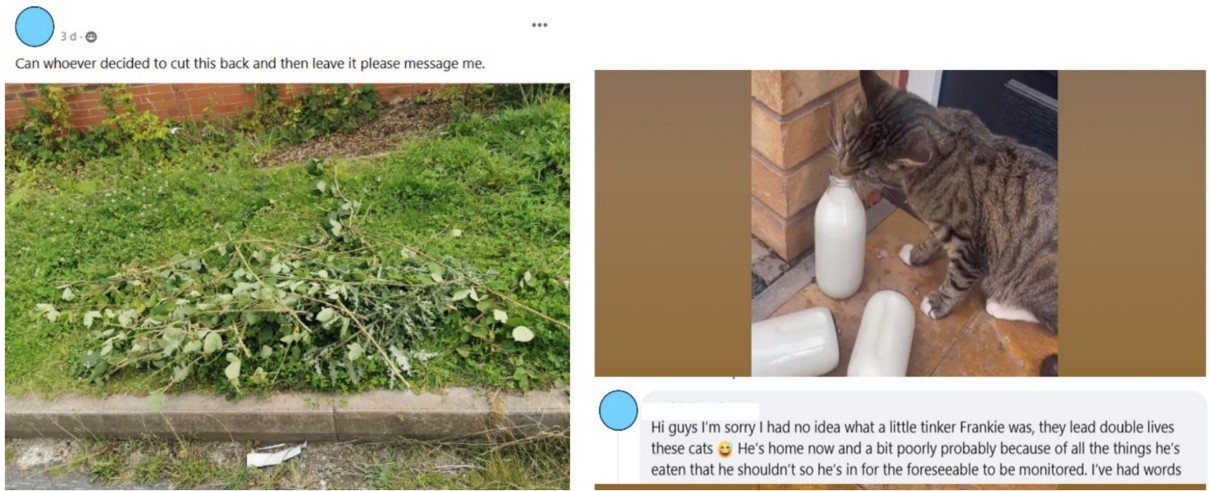

**Figure 5.** Breaching boundaries and 'unruly' human and non-human behaviour as reported in Marston Grange social media exchanges.

This suggests that we might be less sanguine about the prospect of implementing permeable boundary options that extend the plot beyond the 'building envelope and into the landscape' [35]. However, it is also worth bearing in mind the repeated calls for the design of more climate-resilient garden spaces that encourage biodiversity [8], and suggestions from developers, local authorities and residents to deliver positive health, social and environmental benefits in new-build developments. Therefore, Table 1 sets out the possible strengths and weaknesses of different boundary options, based on the authors' assessment of their cost effectiveness, replicability, innovative design, and whether they could be implemented and/or scalable across different contexts.

**Table 1.** Testing different boundary options as 'nature-based solutions'.

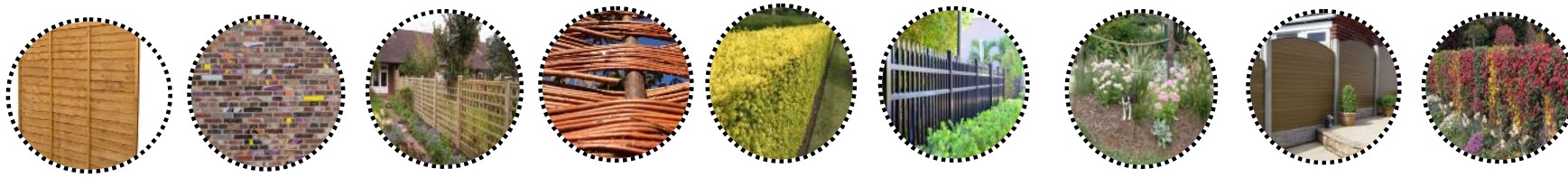

| Possible Nature-Based Solutions (0–5) | "6 ft × 6 ft" Wood Panels and/or Gravel Boards | Walls Created from Local and/or Recycled Materials | Open/Trellis Fencing | Willow/Wicker/ Woven Hurdles | Mono-Species Hedge (Privet, yew, etc) | Metal (Open Design to Encourage Plant Growth/Species Movement) | Rope (Open Design to Encourage Plant Growth/Species Movement) | Recycled/ Composite Fencing | Mixed Species/ Edible Hedges |
|---|---|---|---|---|---|---|---|---|---|
| | | | | | Boundary Type | | | | |
| Wildlife corridors/biodiversity repositories | 1 | 1 | 3 | 2 | 3 | 3 | 3 | 1 | 5 |
| Food and fuel for human/animal use | 2 | 2 | 3 | 3 | 3 | 3 | 3 | 2 | 4 |
| Improving microclimate | 2 | 2 | 3 | 3 | 3 | 3 | 3 | 2 | 4 |
| Providing a cultural link to past/place connection | 2 | 3 | 3 | 3 | 4 | 3 | 3 | 2 | 5 |
| Screening and shading buildings and human activity | 5 | 5 | 4 | 4 | 4 | 3 | 3 | 5 | 3 |
| Reducing pollution, improving air quality | 1 | 1 | 3 | 3 | 3 | 3 | 3 | 1 | 4 |
| Physical boundary (privacy, security and safety) | 5 | 5 | 4 | 4 | 4 | 4 | 2 | 5 | 3 |
| Soil conservation | 2 | 2 | 3 | 3 | 3 | 4 | 4 | 2 | 4 |

**Table 1.** *Cont.*

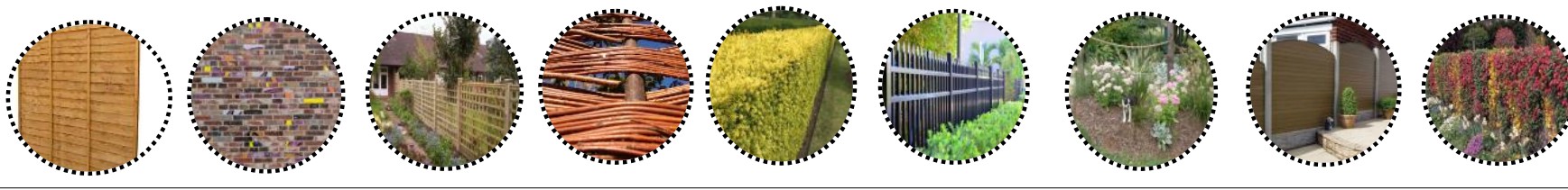

| Possible Nature-Based Solutions (0–5) | *"6 ft × 6 ft" Wood Panels and/or Gravel Boards* | *Walls Created from Local and/or Recycled Materials* | *Open/Trellis Fencing* | *Willow/Wicker/ Woven Hurdles* | *Mono-Species Hedge (Privet, yew, etc)* | *Metal (Open Design to Encourage Plant Growth/Species Movement)* | *Rope (Open Design to Encourage Plant Growth/Species Movement)* | *Recycled/ Composite Fencing* | *Mixed Species/ Edible Hedges* |
|---|---|---|---|---|---|---|---|---|---|
| | | | | | **Boundary Type** | | | | |
| Scalable and replicable adaptation, which can be cost effectively managed | 5 | 5 | 4 | 4 | 4 | 4 | 4 | 5 | 3 |
| Designed to deliver multiple gains over time and across the landscape | 2 | 2 | 4 | 3 | 3 | 3 | 4 | 2 | 4 |
| Create infrastructure that appreciates in value and encourages new skills, innovation and enterprises | 2 | 2 | 3 | 3 | 3 | 3 | 3 | 3 | 4 |
| **TOTAL** | **29** | **30** | **37** | **35** | **37** | **36** | **35** | **30** | **43** |

According to this analysis, fast-growing yield-bearing mixed hedges offer the most benefits (Figure 6). These hedges can provide visually appealing features at different times of year, offering seclusion and shelter, encouraging movement and food resources. Similarly, different hedge species help to dampen noise, and remove dust and other pollutants, creating valuable ecological corridors for xylophagous organisms, amphibians, birds, reptiles and mammals supplying food (berries, leaves, fruits, vegetables, and herbs) and other vegetable matter (fuel, timber, and compost), thus contributing to a potentially sustaining cycle of localized production and consumption [9].

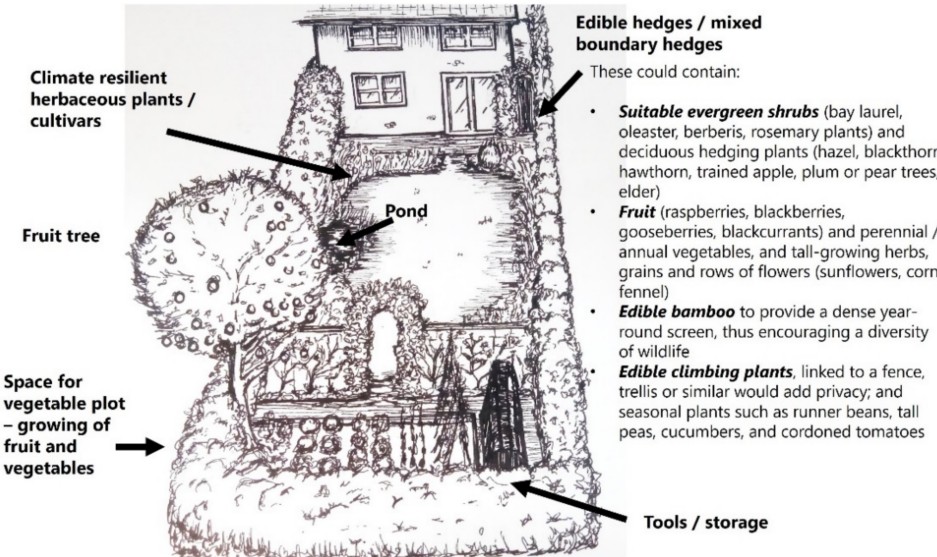

**Figure 6.** A possible design for rear garden space, indicating edible planting and different hedge boundaries. Source: Authors' own sketch.

Advice would have to be sought regarding the cultivation and management of possible combinations of hedge species. The composition and arrangement of plant and tree species would also require careful consideration, particularly regarding the suitability of texture, colour, shape, smell, foliage, height, width in relation to plot dimensions and local conditions. Sunlight, shade, temperatures, wind, soil types, and rainfall are important considerations here [10]. Plants may have extensive root systems that compete for soil nutrients and water. The leaves, twigs and other organic matter from the hedges will decompose to create soil humus, thereby increasing permeability and fertility, and potentially retarding surface water run-off.

Much of the Marston Grange scheme is being built out. Yet, there is scope to reflect on how functional and healthy landscapes could be created using networks of hedgerows and/or other sustainable boundary treatments, reflecting the need for the creation of climate-resilient gardens that "should facilitate the movement of wildlife" [8]. Aside from providing much-needed privacy, this would encourage the transfer of animals and people across property lines, and hedges provide potentially valuable micro-ecosystems for different pollinators and other fauna. These 'hybrid spaces' also carry the potential to generate regular and hopefully positive garden-related social exchanges [6], while increasing landscape connectivity.

## 7. Building Scale Models

Against the wider backdrop studies that call for the testing of possible scalable ecologically connected food production spaces at the contested rural–urban fringe [14,15], Figure 7 represents a reworked example of the Marston Grange scheme. In this case, rather than focusing exclusively on developing infrastructure, buildings, roads and plots, landscape elements and biological components are fundamentally important in early design thinking

for sustainable communities [9]. The overarching ambition in this is for the urban form to be structured in large part by the green (and blue) infrastructure: new hedgerows connect with existing ones, while rear garden hedges combine to create green networks. This responds to criticisms regarding the need for landscaping to respect and enhance biodiversity, connecting the spatial arrangement of new and existing landscape features with historical factors, including field boundaries and farming practices [11].

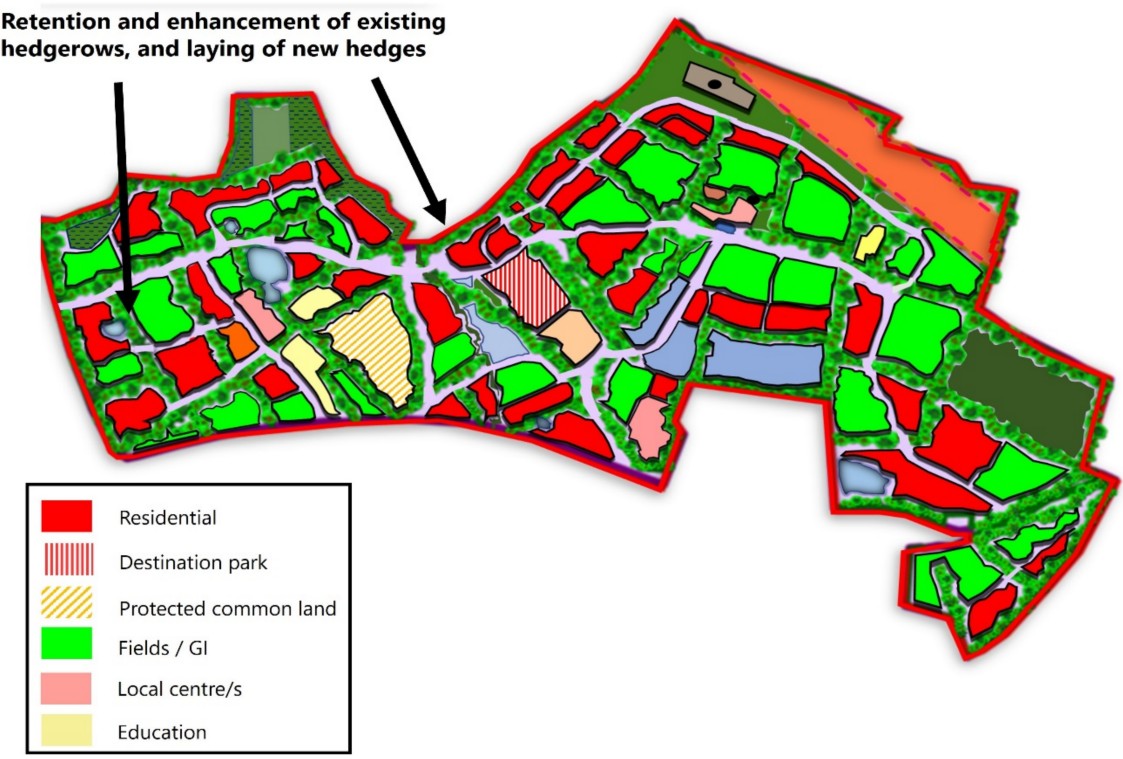

**Figure 7.** A remodelled North of Stafford Masterplan [34].

Initial responsibility for hedge laying and planting could form part of the landscape plan and contract of works agreed by the developer, landowner, local authority and relevant contractors. Developer contributions and/or service charges might be negotiated at the planning decision stage for planting/installation and aftercare arrangements. A small service charge could be paid by residents to those property management companies that often maintain communal areas and shared services on new properties; this charge might logically form part of the sale and be referenced in freehold property deeds/tenancy arrangements. Local authority monitoring would be secured through the planning process via the discharge of landscaping-related planning conditions, including a schedule of works detailing the type of native, mixed-hedge species, planting seasons, maintenance arrangements, and so on. Enforcement relating to the breach of conditions could be addressed locally. There is scope, too, for sophisticated technologies, like those used in some fruit harvesting, to be used in the monitoring of newly created hedge networks [12].

More ambitiously, it is conceivable that new developments might centre around working farms and/or inclusive local growing spaces: resilience and social cohesion are generated through communal growing, thus encouraging a sense of place [30]. With shades of those more 'radical' suburban ideas promulgated by certain British architects, consultant planners and professional officers for housing to be arranged around communal productive green, hedged spaces [19], the starting point here is to recognize the significance of the countryside spaces and natural systems, rather than focus on the layout of buildings, roads, and infrastructure, and the displacement or careful choreographing of protected/marketable species to align with human design ambitions. This also connects with a need to create sus-

tainable and resilient food networks across productive residential landscapes, embedding food systems thinking into planning new and existing landscapes [14].

This reimagined planned community, guided by a proposed hedgerow structure, centres around farm production and/or gardening activity, with varied land uses, including fields and infrastructure set aside for arable and/or pastoral farming, and civic/commercial agriculture. Ultimately, without lapsing into an overly nostalgic rendering of the landscape, this reworked example carries the potential to "maintain the landscape authenticity" [9], increasing aesthetic appeal, building public trust in the design and implementation of large-scale housing schemes, while delivering healthy, affordable food capable of serving diverse populations and resisting future socio-economic crises [16]. Likewise, Figure 8 tentatively sketches out a path for the design and implementation of other developments around working farms. This holds the obvious potential for the creation of sustainably designed buildings.

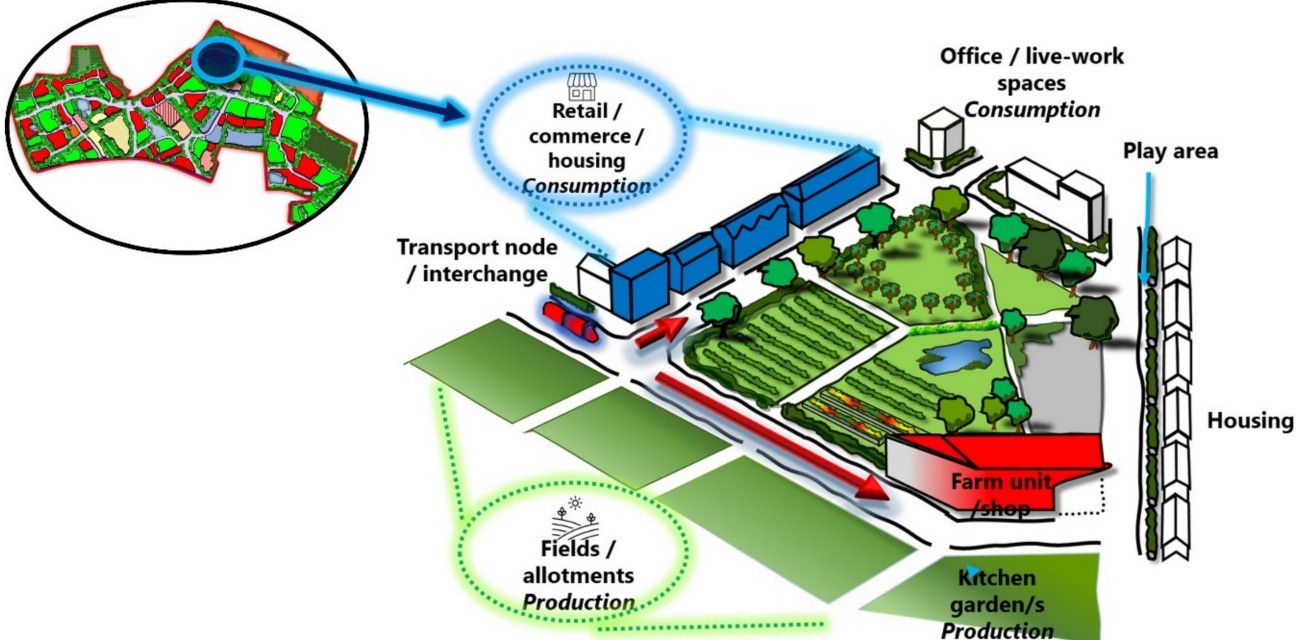

**Figure 8.** Residential development created around existing and new farm activities, helping to create a circuit of production and consumption.

Employment, recreational and educational opportunities are also generated for innovative food growers/producers, and by encouraging new growers, children, young adults and other community stakeholders to socially and ecologically integrate into inclusive, liveable spaces. These could include acquiring new skills and building shared ideals around the nutritional value of adopting shorter, sustainable supply chains, and sustainable local farming [14,15]. The design also encourages a reworked vision of suburban urbanism; and bringing people closer to the psychological and emotional benefits of nature (Figure 9).

Acting as a productive agri-environment scheme capable of delivering environmental public goods, it holds potential to serve local and wider markets (Figure 10). This could form part of wider initiatives to identify suitable official and unsanctioned growing spaces across the urban matrix; this design promises to 'knit together' architectural, design and technological interventions with diverse typologies of (sub)urban spaces.

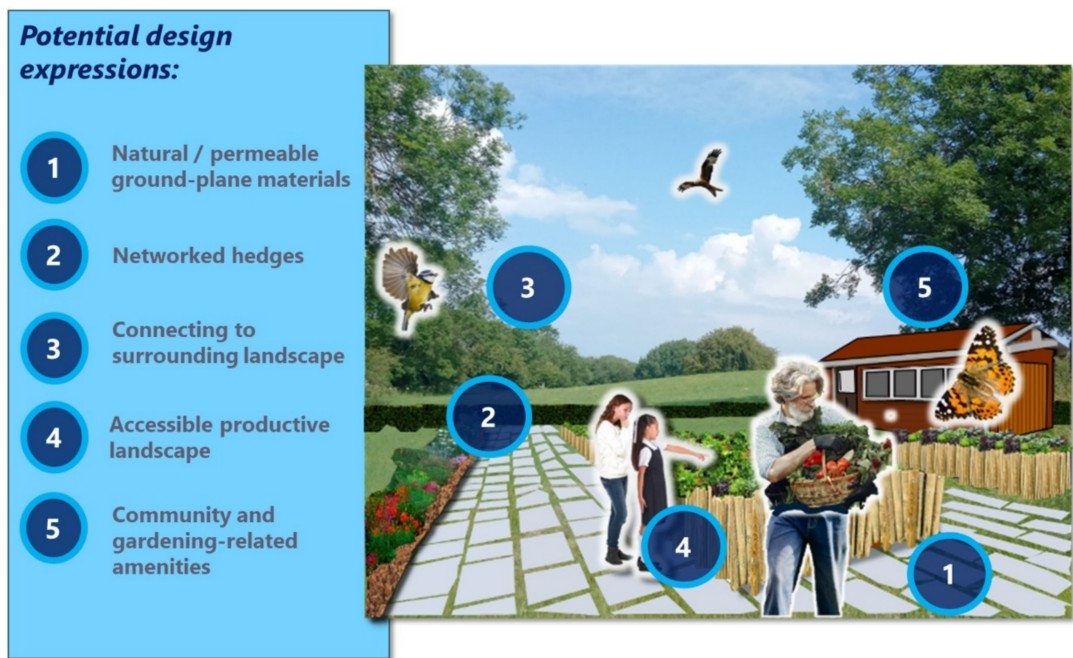

**Figure 9.** A community gardening/kitchen garden space that could form part of an agriculture-led residential development. Source: Adapted from Wulfkuhle (2022) Adapted from [31].

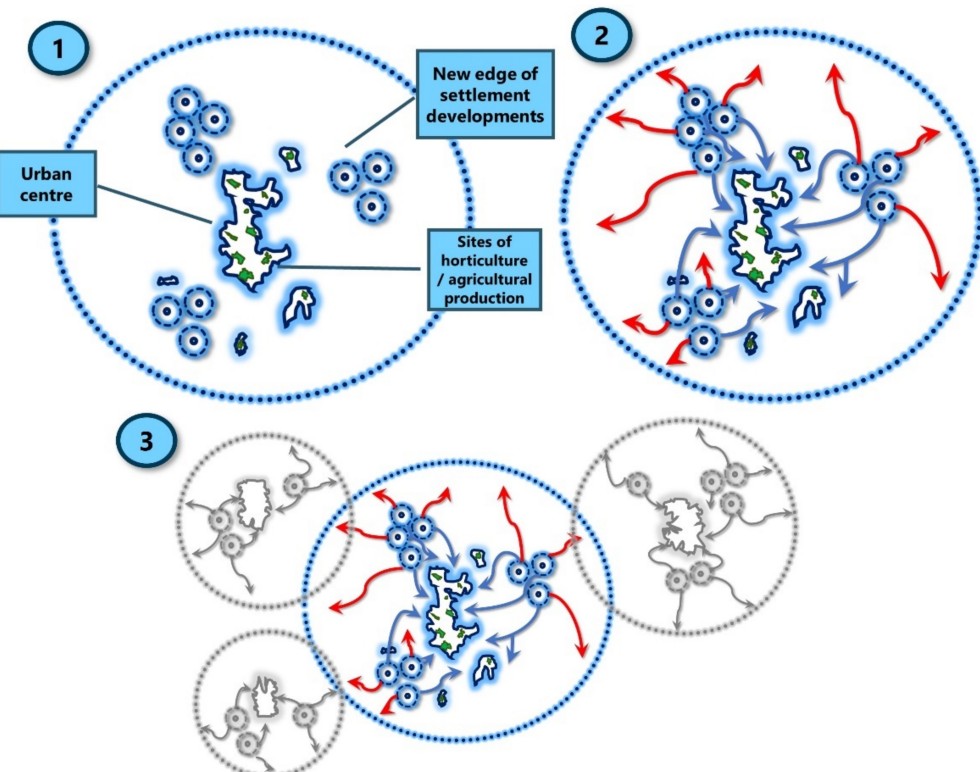

**Figure 10.** The scaling 'up' and 'out' of agricultural-led residential development. This could involve an analysis of existing urban sites capable of supporting agricultural production (1). The building of new edge-of-settlement sites (2); these hold the capacity to grow food and support the main, 'nodal' settlement, while foodstuffs could be 'exported' to nearby urban areas and beyond (2). A network of inter-connected food-growing urban areas (3).

## 8. Challenges

Many practical challenges surround the possible implementation of these ambitions. The realization of possible benefits depends in large part on the acquisition of land. One sweeping option could involve the purchasing of land at prices as close to agricultural value as possible; any uplift in value generated from the development is then captured locally and re-invested in local physical, social and environmental services and amenities. This approach could involve the use of acquisition powers like those applied in the creation of British government-sponsored post-war New Town Corporations. Alternatively, given recent decades of shrinking public finances, resource and pecuniary constraints, and deregulation of private enterprise, a more moderate approach might involve a repurposed private-developer consortium-type arrangement, akin to those launched in the 1980s to build 'new country towns'—with little success [41]. Arguably, the most politically palatable and expedient model could involve the use of 'reinvigorated' spatial planning instruments that built around a stronger ecologically inspired 'caring for place' [4]. From here, criteria can be created relating to scale, contribution to housing need, local support, commitment to quality, and consideration of infrastructure. Establishing interest in the possibilities of development could involve local authorities and relevant planning advisory services engaging in 'dynamic and ongoing' discussions between landowners, farmers, residents, and other stakeholders [14]. This dialogue would include a (re)consideration of site suitability, as some such sites are likely to be allocated for development in existing statutory planning frameworks.

Such a model would not necessarily result in the swift delivery of housing currently coveted by politicians, investors, developers and some potential occupiers. Shifting public opinion and expectations may also be challenging; some residents may not wish to be associated with 'green' activities, food production and the potentially unsettling sights, smells and sounds of agriculture in such proximity to residential areas [2]. Potential investors may also be dissuaded by the image associated with this lifestyle; and some developers would fear creating agriculture-based development because it diverges too far from established modes of practice [31]. While this may be the case, evidence from recent US, Canada, Europe and elsewhere suggests that marketability of development is enhanced among those individuals seeking a closer connection to local food growing initiatives; and promising policy initiatives are helping shift entrenched stakeholder views on the possibilities of food growing at the urban fringe [29–31]. This does raise obvious concerns over how these schemes use nature in ways to upscale development, thereby diluting those more community-spirited, ecological ideas. Thorny questions of ownership also emerge; homeowners and landlords may hold titles to their property, though farmland and community growing spaces could be developer-owned and operated, to full nonprofit-owned and/or leasehold [30]. Working towards a socially cohesive vision that accounts for different tastes, and possible ownership complexities requires time, effort and resources [14].

In some cases, early initiative has been taken via the creation of a non-profit entity with a board of directors constituted by community members and key stakeholders [30,31]. Meaningful and sustained engagement is then developed with landowners, local planners, education providers, residents, and community groups. A Memorandum of Understanding or similar is one obvious route to delineate the responsibilities; this coalition of actors is then responsible for coordinating the management, networking and resource capture [30]. The skills of land agents, architects, planners and lending institutions would be needed to navigate local planning processes and regulatory frameworks. Similar negotiations with possible developers and service providers, regarding development phasing and the supply of water, electricity, gas, and waste disposal would also have to take place ahead of development.

Typically, though not exclusively, such developments operate under the purview of community/homeowners' associations, though professional services are required to work through construction-related costings, budgets for operating, maintenance and administration, and the scale of operation (type of crop production, for example). Under this

arrangement, food procurement strategies, food education and purchasing systems with public agencies (local authorities, for example) can emerge, incorporating sectors involved with the distribution, processing, marketing and consumption of food [32]. Management and business plans would be needed for farm-related activities, including accounting for projected revenue streams, staffing, labour, and ongoing costs [30].

Lessons can also be learned here from broader established urban food-growing schemes, which raise awareness around the need for wider networks to sustain activities. Such networks often enable knowledge sharing and funding support, linking through to the rise in urban food policies may also enable more support for schemes and the scaling-up of these solutions, weaving together other urban food growing spaces and forming part of a movement to create productive landscapes.

## 9. Concluding Thoughts

This study has explored how established nature–culture binaries attached to traditional models of large-scale edge-of-settlement development, as reflected in standardized landscaping arrangements, plot design and boundary treatments, are challenged by different human and non-human interactions. It represents an important step towards moving the focus of away from profits, quantity and economic exchange value and human territoriality traditionally associated with deliberative 'greening' efforts used in the design and marketing of new suburban developments. Instead, emphasis is placed on outlining how hedges and/or other porous designs based around a deeper "consideration for more-than-human residents" [3] might increase ecological connectivity, build climate-resilient gardens, and encourage sociality, especially at the early stages of the design process for large-scale residential development.

Connecting to and extending recent ideas around the role and function of embedding urban hedgerows into official urban planning processes, the reworked Marston Grange scheme is based more around the existing ecological and landscape qualities. Based around hedged field boundaries, this reimaging proposal incorporates a network of existing and newly planted hedgerows to structure neighbourhood design; hence, this moves towards creating integrated urban food systems, rather than isolated, piecemeal opportunities for community gardening. Instead, the design outlined here would maintain and protect biodiversity, establish a deeper human connection with local history, culture and ecology, and encourage forms of residential development centred around existing and/or improved agricultural initiatives which could form part of a wider sustainable food system [16]. Thus, the opportunity is also there to challenge existing thinking, outlining one possible model in the wider pursuit of creating stronger policies and models of delivery applicable to other peri-urban contexts.

One logical step would involve drawing on the experience of those human actors who would have a stake in the design and implementation of agricultural-focused forms of residential development. This evidence would further highlight some of the challenges and opportunities associated with building wider urban food networks, through connecting spaces, policy integration and support to sustain activities.

**Author Contributions:** Conceptualization, D.A., P.J.L. and M.H.; Methodology, D.A., P.J.L. and M.H.; Formal analysis, D.A., P.J.L. and M.H.; Investigation, D.A., P.J.L. and M.H.; Data curation, D.A.; Writing—original draft, D.A., P.J.L. and M.H.; Writing—review & editing, D.A., P.J.L. and M.H.; Visualization, D.A. All authors have read and agreed to the published version of the manuscript.

**Funding:** This research received no external funding.

**Data Availability Statement:** The data used to support the findings of this study can be made available by the corresponding author upon request.

**Conflicts of Interest:** The authors declare no conflict of interest.

## Notes

1   Countryside hedges in England are statutorily protected according to their length, location and importance: https://treecouncil.org.uk/what-we-do/hedgerows/ (accessed on 9 September 2023).

2   In England, there is a mandatory biodiversity net gain requirement of at least 10% for new developments from 2023/4, while some leading house builders pledge to identify planting opportunities that increase flora and fauna.

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
