# Peer review of "Edible Garden Cities: Rethinking Boundaries and Integrating Hedges into Scalable Urban Food Systems"

_land, doi:10.3390/land12101915_

Round 1

Reviewer 1 Report (New Reviewer)

Dear authors

Thank you very much for this interesting paper. I enjoyed reading it. 

I believe that the manuscript can be published as it is. 

Author Response

Thank you to the reviewer for the kind words and comments.  An improved version of the paper is being prepared.

Reviewer 2 Report (New Reviewer)

The problem of fencing with 'hard' landscaping (a full fence brick or masonry) is an affliction not only English. For this reason, I find the text interesting and possible to implement also in other cultural circles, even where there is not such a rich tradition of shaping hedges as in England. 

It is worth enriching Chapter 8 with a sample synoptic diagram showing the implementation (at different levels / scales) of concepts related to the introduction of plant partitions / hedgerows, taking into account the issue of title food systems.

Line 489-491 (Figure 6) - no explanation (legend) for the markings on the materplan

Author Response

The section on implementation has now been bolstered; the improvements are highlighted in yellow.  A new figure (Figure 6) shows possible indicative planting arrangements and design expression / implementation. Additionally, Figure 6 has been improved.

Reviewer 3 Report (New Reviewer)

The Abstract of this paper does not exactly represent the intent & content. The Title implies food systems can be incorporated into new large-scale peri-urban residential developments through innovative boundaries and hedges. The abstract states the paper explores ‘porous boundary treatment and plot layouts;’ but the boundary treatments are represented in one table only and plot layouts not at all. Although the referencing includes much of the recent work on hedgerows & planning, it does not include the extensive published work since 2000 on new models for peri-urban residential/agricultural developments such as described by the contributors to James Corner’s ‘Recovering Landscape’, or de Geyter’s ‘Aftersprawl’(2002) or in 2003’s ‘Event Landscape’, where the innovative edge residential/rural developments in Germany are  described.

Specifically, the Introduction needs to define the main terms and give examples, as well as including windbreaks with hedgerows.  Suburbanising Nature covers interesting points.  Boundary Marking needs to be retitled, for example Ecological, Biodiverse & More-than-human Issues to be more in keeping with the Abstract. Growing Ambitions provides the location for discussion on all the work already done on peri-urban/agric-res developments; such as lines 276-286.  Methods & Approaches appears to provide the scientific rigour required.  In terms of the ‘secondary material’ review, it should include the relevant last decade of peri-urban models. Pushing Boundaries addresses the habitat potential highlights & how enhancing low-grade agricultural land is not happening.  Table 1 is informative.  Suggest that lines 455-463 describing possible edible planting scenarios could also be presented in tabular/diagrammatic form.  Building Scale Models again needs to refer to recent peri-urban models.  Figure 6 is meaningless without a legend, scale, sun/wind etc.  Figure 8 does not provide enough to show ‘a functioning agriculture-led residential development’.  Lines 455-468 & 499-504 need supporting figures to show possible hedgerow profiles according to certain plantings.  The height and width variations should also be indicated and the cadastral implications.  Challenges are only partially acknowledged and the ‘precedence analyses’ line582 does not provide enough recognition of precedents and how hedgerow networks are different model of productive land. Lines 602-610 show that the authors do not clearly distinguish between peri-urban ag crops and the particularities of hedgerow agriculture e.g. line605 suggests livestock.

In summary, the paper needs to be reworked to address the title more clearly.

Figure Conventions and Grammatical issues  

Figures need dates as well as source. Fig 2 caption L303 ‘North’.  Fig 4 needs more detailed caption. Fig 6 is incomplete.

L 31 is not grammatically correct

L 41 delete ‘of’

L 44…examines ‘how’…

L 112 ‘plant’ species…

L292 delete ‘was undertaken’

L515 …food systems thinking (in) the planning….

L 622 …large-scale residential [scales] ‘projects’

Suggested references

Aufmkolk, G ‘cet-o/kunst+Herbert “Mississippi“ in Hoppenstedt, Adrian (ed),(2003) EVENT LANDSCAPE, Berlin: Birkhauser pp 98-103.

James Corner (ed) (1999) Recovering Landscapes, NY: Princeton Architectural Press.

De Geyter & Bekaert, G. (2002) After-sprawl: Research for the Contemporary City, Rotterdam: Nai Publishers.

Figure Conventions and Grammatical issues  

Figures need dates as well as source. Fig 2 caption L303 ‘North’.  Fig 4 needs more detailed caption. Fig 6 is incomplete.

L 31 is not grammatically correct

L 41 delete ‘of’

L 44…examines ‘how’…

L 112 ‘plant’ species…

L292 delete ‘was undertaken’

L515 …food systems thinking (in) the planning….

L 622 …large-scale residential [scales] ‘projects’

Author Response

Reviewer comments

The Abstract of this paper does not exactly represent the intent & content. The Title implies food systems can be incorporated into new large-scale peri-urban residential developments through innovative boundaries and hedges. The abstract states the paper explores ‘porous boundary treatment and plot layouts;’ but the boundary treatments are represented in one table only and plot layouts not at all.

Authors’ response: The abstract has been improved to reflect the tighter focus (highlighted in yellow). Hence the abstract, introduction and parts of the literature review and analysis have been improved to reflect this point.  A new figure detailing the possible indicative plot layout is included (Figure 6).

Reviewer comments

Although the referencing includes much of the recent work on hedgerows & planning, it does not include the extensive published work since 2000 on new models for peri-urban residential/agricultural developments such as described by the contributors to James Corner’s ‘Recovering Landscape’, or de Geyter’s ‘Aftersprawl’(2002) or in 2003’s ‘Event Landscape’, where the innovative edge residential/rural developments in Germany are  described.

Authors’ response: The introduction and literature review now include more explicit reference to recent work on the RUF development pressures, and those ongoing studies that seek to strike a balance between urbanisation and preservation / enhancement of the rural hinterland.  These improvements are highlighted in yellow in the main text; and the ideas are drawn from the following sources:

  1. Armstrong, H.; Lopes, A. Re-Ruralising the Urban Edge: Lessons from Europe, USA & the Global South in Balanced Urban Development: Options and Strategies for Liveable Cities; Maheshwari, B.; Thoradeniya, B; Singh, V.P., Eds. Springer Open: Water Science and Technology Library., 2016; pp. 17-28.
  2. De Waegemaeker, J.; Primdahl, J.; Vanempten, E.; Søderkvist, L.; Kristensen, E. R.; Vejre, H. Eur. Plg. Stu. 2023, 31(10), 2235-2253.
  3. Nasr, J.; Potteiger, M. Spaces, systems and infrastructures: From founding visions to emerging approaches for the productive urban landscape. Land 2023, 12(2), 410.

Reviewer comments

Specifically, the Introduction needs to define the main terms and give examples, as well as including windbreaks with hedgerows.  Suburbanising Nature covers interesting points.  

Authors’ response: The introduction (lines 46 – 53) is now bolstered to reflect this point:

In this context, several studies recommend the use of mixed-species hedges in residential contexts [9,10,11,12,13].  These can provide ‘natural’ pest control, shelter, food, carbon storage, infiltration p1romotion, soil nutrients, and increase insect pollinator and invertebrate diversity [9,10,11].1 Hedges also act as air pollution barrier and windbreaks; and can mitigate issues around absorbing / reducing particulate matter, while also creating aesthetically pleasing boundaries for food growing [9, 13]

Reviewer comments

Boundary Marking needs to be retitled, for example Ecological, Biodiverse & More-than-human Issues to be more in keeping with the Abstract.

Authors’ response: This section has been retitled accordingly.

Reviewer comments

Growing Ambitions provides the location for discussion on all the work already done on peri-urban/agriculture-led developments; such as lines 276-286.  

Authors’ response: This section has been reworked; improvements shown in yellow (lines 244-247).  More explicit reference is made to relevant RUF-related studies on creating balance between development and agriculture (again, highlighted in yellow).

Reviewer comments

Methods & Approaches appears to provide the scientific rigour required.  In terms of the ‘secondary material’ review, it should include the relevant last decade of peri-urban models.

Authors’ response: A more explicit reference is made to the use of a case study approach.  Likewise, mention is given to the review of relevant RUF studies (lines 260-263).

Reviewer comments

Pushing Boundaries addresses the habitat potential highlights & how enhancing low-grade agricultural land is not happening.  Table 1 is informative.  Suggest that lines 455-463 describing possible edible planting scenarios could also be presented in tabular/diagrammatic form.  

Authors’ response: A new figure (Figure 6) has been created; this provides a graphic illustration of possible planting schemes that could be embedded into early design thinking.

Reviewer comments

Building Scale Models again needs to refer to recent peri-urban models.  

Authors’ response: Reference is made to relevant RUF material; this includes drawing more on references 14, 15 and 16 to build arguments.  These improvements are highlighted in yellow.

Figure 6 is meaningless without a legend, scale, sun/wind etc.  

Authors’ response: Figure 6, now figure 7 has been improved.

Reviewer comments

Figure 8 does not provide enough to show ‘a functioning agriculture-led residential development’.  

Authors’ response: The title has been amended to better reflect the image: A community gardening / kitchen garden space that could form part of an agriculture-led residential development. Source: Adapted from Wulfkuhle (2022).

Reviewer comments

Lines 455-468 & 499-504 need supporting figures to show possible hedgerow profiles according to certain plantings.  The height and width variations should also be indicated and the cadastral implications.  

Authors’ response: Figure 6 should give a sense of certain possible plantings.  Lines 463 – 465 have been amended to reflect the need for more practical advice regarding suitable planting arrangements, including a comment on height / width.

Reviewer comments

Challenges are only partially acknowledged and the ‘precedence analyses’ line582 does not provide enough recognition of precedents and how hedgerow networks are different model of productive land.

Authors’ response: The Challenges section sets the idea of hedged residential developments at the RUF in a wider context.  The section now better reflects the other RUF-related work that details the lessons and pitfalls of implementation, and how these might play out in an English planning context.

Reviewer comments

Lines 602-610 show that the authors do not clearly distinguish between peri-urban ag crops and the particularities of hedgerow agriculture e.g. line605 suggests livestock.

Authors’ response: Reference to livestock has been removed.

Reviewer comments - Figure Conventions and Grammatical issues  

Figures need dates as well as source. Fig 2 caption L303 ‘North’.  Fig 4 needs more detailed caption. Fig 6 is incomplete.

L 31 is not grammatically correct

Authors’ response: This has been altered.

L 41 delete ‘of’

Authors’ response: Deleted

L 44…examines ‘how’…

Authors’ response: How inserted

L 112 ‘plant’ species…

Authors’ response: Amended

L292 delete ‘was undertaken’

Authors’ response: Deleted

L515 …food systems thinking (in) the planning….

Authors’ response: Amended

L 622 …large-scale residential [scales] ‘projects’

Authors’ response: Amended

Reviewer 4 Report (New Reviewer)

This paper takes an interesting more-than-human approach to examine a possibility of scalable urban food systems while relaxing human-nature tensions.

Conceptually, this is an interesting paper yet methodologically, the elaboration on data generation process can be improved to authenticate data provided by this empirical case study. For instance, who have rated these different boundary types in Table 1?

Although the idea of reimagining a planned community is innovative, how to implement is still very abstract and idealistic. For instance, how to distribute “safe, healthy, affordable food” produced by the proposed hedgerow structure? Who can have access to these foods and who is in charge of growing these foods? How to differentiate civic versus commercial agriculture? How inclusive are these invented food productions related spaces?

If the development is an agriculture-based development, who are the food growers/farmers, especially there is a plan to upscale.

Is there existing public policy to regulate these new development? 

No comment. 

Author Response

Reviewer comments

This paper takes an interesting more-than-human approach to examine a possibility of scalable urban food systems while relaxing human-nature tensions. 

Conceptually, this is an interesting paper yet methodologically, the elaboration on data generation process can be improved to authenticate data provided by this empirical case study. For instance, who have rated these different boundary types in Table 1?

Authors’ response:  This point is explained in the Methods (lines 317- 321; highlighted in yellow) and reaffirmed in line 443.

Although the idea of reimagining a planned community is innovative, how to implement is still very abstract and i)dealistic. For instance, how to distribute “safe, healthy, affordable food” produced by the proposed hedgerow structure? Who can have access to these foods and who is in charge of growing these foods? How to differentiate civic versus commercial agriculture? How inclusive are these invented food productions related spaces? If the development is an agriculture-based development, who are the food growers/farmers, especially there is a plan to upscale. Is there existing public policy to regulate these new development?

Authors’ response:  The implementation remains tentative; this is deliberate, given that a scheme of this kind would be represent a radical move for English land use planning.  That said, more explicit attention is given to the lessons from other similar cases at the RUF in other contexts – the ‘Challenges’ section provides details of land ownership, planning instruments and policy initiatives needed to initiate a change of thinking at the early design stage.  More is made of who would oversee the design and implementation and other practical matters.  Improvements are highlighted in yellow.

Round 2

Reviewer 3 Report (New Reviewer)

The revisions address all the reviewer's comments

This manuscript is a resubmission of an earlier submission. The following is a list of the peer review reports and author responses from that submission.

Round 1

Reviewer 1 Report

This paper is well presentation, but I do not get it is an review or a research article. As a research article, this paper lacks the scientific gap. Following comments should be improved.

(1)  What is the scale for the growing the garden? The garden is a community garedn, or is a urban park.

(2) Figure 2 and figure 3 could be merged.

(3) I did get the meaning of figure 4. Do you want to compare the changing remote sensing images?